# Unmasking the Risk Factors Associated with Undiagnosed Diabetes and Prediabetes in Ghana: Insights from Cardiometabolic Risk (CarMeR) Study-APTI Project

**DOI:** 10.3390/ijerph21070836

**Published:** 2024-06-26

**Authors:** Thomas Hormenu, Iddrisu Salifu, Juliet Elikem Paku, Eric Awlime-Ableh, Ebenezer Oduro Antiri, Augustine Mac-Hubert Gabla, Rudolf Aaron Arthur, Benjamin Nyane, Samuel Amoah, Cecil Banson, James Kojo Prah

**Affiliations:** 1Department of Health, Physical Education and Recreation, Faculty of Science Technology Education, College of Education Studies, University of Cape Coast, Cape Coast 00233, Ghana; juliet.paku001@stu.ucc.edu.gh (J.E.P.); eric.awlime-ableh@stu.ucc.edu.gh (E.A.-A.); ebenezer.antiri@stu.ucc.edu.gh (E.O.A.); augustine.gabla@stu.ucc.edu.gh (A.M.-H.G.); 2Cardiometabolic Epidemiology Research Laboratory, Department of Health, Physical Education and Recreation, University of Cape Coast, Cape Coast 00233, Ghana; iddrisu.salifu@stu.ucc.edu.gh (I.S.); rudolf.arthur@ucc.edu.gh (R.A.A.); benjamin.nyane@ucc.edu.gh (B.N.); 3Directorate of University Health Services, University of Cape Coast, Cape Coast 00233, Ghana; samuel.amoah@ucc.edu.gh (S.A.); cecil.banson@ucc.edu.gh (C.B.); james.prah@ucc.edu.gh (J.K.P.)

**Keywords:** fasting plasma glucose, generalized structural equation model, Ghana, risk factors, undiagnosed diabetes

## Abstract

Introduction: Undiagnosed diabetes poses significant public health challenges in Ghana. Numerous factors may influence the prevalence of undiagnosed diabetes among adults, and therefore, using a model that takes into account the intricate network of these relationships should be considered. Our goal was to evaluate fasting plasma levels, a critical indicator of diabetes, and the associated direct and indirect associated or protective factors. Methods: This research employed a cross-sectional survey to sample 1200 adults aged 25–70 years who perceived themselves as healthy and had not been previously diagnosed with diabetes from 13 indigenous communities within the Cape Coast Metropolis, Ghana. Diabetes was diagnosed based on the American Diabetes Association (ADA) criteria for fasting plasma glucose, and lipid profiles were determined using Mindray equipment (August 2022, China). A stepwise WHO questionnaire was used to collect data on sociodemographic and lifestyle variables. We analyzed the associations among the exogenous, mediating, and endogenous variables using a generalized structural equation model (GSEM). Results: Overall, the prevalence of prediabetes and diabetes in the Cape Coast Metropolis was found to be 14.2% and 3.84%, respectively. In the sex domain, females had a higher prevalence of prediabetes (15.33%) and diabetes (5.15%) than males (12.62% and 1.24%, respectively). Rural areas had the highest prevalence, followed by peri-urban areas, whereas urban areas had the lowest prevalence. In the GSEM results, we found that body mass index (BMI), triglycerides (TG), systolic blood pressure (SBP), gamma-glutamyl transferase (GGT), and female sex were direct predictive factors for prediabetes and diabetes, based on fasting plasma glucose (FPG) levels. Indirect factors influencing diabetes and prediabetes through waist circumference (WC) included childhood overweight status, family history, age 35–55 and 56–70, and moderate and high socioeconomic status. High density lipoprotein (HDL) cholesterol, childhood overweight, low physical activity, female sex, moderate and high socioeconomic status, and market trading were also associated with high BMI, indirectly influencing prediabetes and diabetes. Total cholesterol, increased TG levels, WC, age, low physical activity, and rural dwellers were identified as indirectly associated factors with prediabetes and diabetes through SBP. Religion, male sex, and alcohol consumption were identified as predictive factors for GGT, indirectly influencing prediabetes and diabetes. Conclusions: Diabetes in indigenous communities is directly influenced by blood lipid, BMI, SBP, and alcohol levels. Childhood obesity, physical inactivity, sex, socioeconomic status, and family history could indirectly influence diabetes development. These findings offer valuable insights for policymakers and health-sector stakeholders, enabling them to understand the factors associated with diabetes development and implement necessary public health interventions and personalized care strategies for prevention and management in Ghana.

## 1. Introduction

Undiagnosed diabetes mellitus, a global health issue, contributes to morbidity and mortality [1]. The international diabetes federation (IDF) estimates that more than half of individuals with diabetes worldwide are unaware of their condition [2]. The rise in undiagnosed impaired glucose tolerance has fueled the increasing prevalence of diabetes mellitus [3]. In Ghana, the increasing diabetes burden exacerbated by COVID-19 poses challenges to the public health and healthcare systems, with many cases remaining undiagnosed, leading to delayed intervention and complications. This study aimed to identify the risk factors for undiagnosed diabetes and prediabetes in Ghana to inform targeted interventions and public health strategies.

Diabetes poses a significant challenge globally, particularly in sub-Saharan Africa, where it affects the region disproportionately [2]. The COVID-19 pandemic has exacerbated this situation, as many deaths in the region are associated with underlying health conditions such as diabetes and cardiovascular disease risk factors. Understanding the causes of diabetes, including beta cell failure and insulin resistance, is essential for developing effective interventions to prevent and treat this condition [4]. However, there is a lack of accurate data on the prevalence of diabetes and other cardiovascular diseases (CVD) risks in sub-Saharan Africa [5]. This lack of information hinders efforts to prepare for and address the impact of future pandemics on people living with the disease.

Ghana, like many other low- and middle-income countries, is experiencing a diabetes epidemic due to demographic shifts, urbanization, and lifestyle changes. The prevalence of diabetes in Ghana has steadily increased from 1.6% in 1974 to 6.3% in 2010 [6], and the number of undiagnosed cases has been significant. The burden of diabetes extends beyond diagnosed cases, as the number of undiagnosed individuals in developing countries, such as Ghana and Sub-Saharan Africa, is substantial. As Osei-Yeboah et al. [7] noted, preventive measures are urgently required to address this increasing trend in Ghana. However, the extent of undiagnosed diabetes and prediabetes status and the factors causing these remain a critical knowledge gap in the Ghanaian context, hindering effective public health responses.

Several factors have been suggested to increase the risk of diabetes [5]. To design effective interventions, it is crucial to understand these risk factors. Early detection is critical to prevent diabetes-related complications, such as cardiovascular disease, neuropathy, retinopathy, and other debilitating conditions [8]. Undiagnosed diabetes perpetuates the epidemic by allowing individuals to continue unhealthy behaviors, unknowingly worsening their condition, and spreading unhealthy habits within communities [1]. To break this cycle, a deeper understanding of the factors contributing to undiagnosed diabetes is needed. Identifying at-risk individuals in indigenous communities can help manage the disease and reduce its impact on the healthcare system. Moreover, the economic consequences of undiagnosed diabetes are substantial, as the cost of managing complications exceeds that of early detection and intervention. Addressing the undiagnosed population not only improves health outcomes, but also alleviates the economic burden on individuals, families, and the healthcare system.

Ghana has epidemiologically transitioned to a more urbanized and sedentary lifestyle, along with changes in diet such as eating high energy dense foods, sweets, and fast foods, contributing to the increase in diabetes and prediabetes cases [9]. Moreover, the interplay between traditional and modern lifestyles in Ghana creates unique risk factors for diabetes, which must be thoroughly explored [10]. We argue that access to healthcare and diagnostic services as well as cultural beliefs varies across regions and all these may affect individuals’ likelihood of being tested for diabetes. In addition, socioeconomic factors, educational level, and health literacy also play crucial roles in managing diabetes, and understanding their impact on the undiagnosed population cannot be underestimated.

Despite prior studies on this topic, undiagnosed diabetes and prediabetes in Ghana remain a significant public health concern [6,7]. This study aimed to identify the direct and indirect factors associated with these conditions and to address the gap in our understanding. A generalized structural equation model (GSEM) was utilized to investigate individual, societal, environmental, and healthcare system factors that contribute to undiagnosed diabetes. The findings provide recommendations for targeted interventions and public health strategies. Addressing the unique challenges faced by the Ghanaian population can lead to improved health outcomes and a reduced burden of diabetes in the country. This study contributes to the global discourse on diabetes prevention and management.

## 2. Methods

### 2.1. Research Design

The cardiometabolic metabolic risk (CarMeR) study is a community-based cross-sectional study conducted using the WHO-STEPwise approach to identify and explain factors associated with cardiometabolic diseases.

### 2.2. Population

The population estimate for the Cape Coast Metropolis was 102,533 people [11]. The CarMeR study included adults aged 25–70 from households in the Cape Coast metropolitan area. This age group was selected because it has a higher risk of developing diabetes in urban areas [12]. The study included participants from both rural and urban communities in metropolitan areas. The diverse nature of the Cape Coast population led to the selection of participants based on specific criteria, such as the type of community they reside in (rural, peri-urban, or urban), their main economic activity (agriculture, fishing, trade, formal employment), dominant religion (Christianity, Islam, African traditional religion), and main ethnicity (Fante, Ewe, Asante) [12].

### 2.3. Participants Recruitment and Selection for the Study

The population was large and geographically dispersed; hence, the multistage sampling technique was employed to recruit 1200 self-identified healthy individuals from 13 urban and rural communities who had not received a diagnosis of diabetes or prediabetes prior to the study to collect baseline data. This was performed in three stages. The first stage involved dividing the population into clusters based on the existing communities within the metropolis. The second stage involved random sampling of the clusters used in the study. At this stage, 13 clusters representing rural, peri-urban, and urban communities with specific characteristics were selected. The third stage involved further sampling within each selected cluster. This involved randomly selecting households and recruiting the first two persons from each family bloodline (with random replacement) in indigenous communities and convenience sampling for non-indigenous communities. Approximately 90–100 participants were randomly selected from three rural communities, four fishing communities, three peri-urban communities, one urban community (university community), two market communities, and one artisanal community. 

### 2.4. Inclusion Criteria

The study included men and women aged 25–70 years who had not been diagnosed with prediabetes or diabetes and were willing to undergo the screening procedure after (8 h) of fasting prior to glucose testing, BMI assessment, lipid level measurement, and body composition analysis.

### 2.5. Exclusion Criteria

The CarMeR study excluded certain individuals, including pregnant women, breastfeeding mothers, those with pre-existing diabetes (type 1 or type 2), those with a BMI below 18.5 kg/m^2^, those with anemia, and those taking high-cholesterol medications as well as any glycemic reducing medicines like metformin, glucocorticoids, and thiazide diuretics. However, those on antihypertensive medications were included in the study. 

These criteria were chosen to align with the study’s primary goal of determining the prevalence of undiagnosed diabetes in the target population. Pregnant women and those with existing diabetes were excluded to maintain the focus of the study on detecting undiagnosed cases, whereas those with a low BMI or anemia were excluded to ensure the validity of the results.

### 2.6. Data Collection Procedures

The CarMeR study was primarily conducted in community centers, including the cardiometabolic epidemiology research laboratory (CERL) and mosques. Permission was obtained from community leaders such as traditional chiefs, opinion leaders, imams, and assembly members. Ethical clearance was granted by the University of Cape Coast Institutional Review Board (UCCIRB/EXT/2022/27). Data were collected from two participants per household, with participants aged 25–70 years selected randomly. Participants were informed of the study’s purpose, benefits, and risks. Written and verbal consent was obtained for each part of the study, including survey questionnaires, anthropometric measurements, and biochemical sample collection.

### 2.7. Questionnaire/Survey Questionnaire

Data collection for the CarMeR study included survey questionnaires, anthropometric measurements, and biochemical sample collection. The WHO-STEP-wise approach was used to identify and explain the risk factors for cardiometabolic diseases. The WHO STEP-wise questionnaire was adapted and modified for the Ghanaian non communicable diseases (NCDs) context following a previous study on NCDs in Ghana [13]. The first step involved gathering demographic and behavioral risk factor data using a questionnaire, while the second step involved collecting physical measurements. The third step involved the collection of fasting blood samples for biochemical tests. The survey was composed of seven validated questionnaires adapted to fit Ghanaian culture and lifestyles. The questionnaires collected data on physical activity, dietary intake, and participant demographics.

### 2.8. Anthropometric Measurements 

Physical measurements, including height and waist circumference, were obtained using equipment validated by international bodies. A stadiometer and tape measure were used for anthropometric measurements. The GS6.5B Body Composition Analyzer (Electro Health Systems, Gujurat, India) was used to measure body weight and waist-to-hip ratio, and bioimpedance analyses were conducted to assess total body fat, metabolic age, and muscle mass. The device calculates various body parameters based on the impedance values. The bioimpedance analysis results were displayed on a screen and printed. Vital blood pressure statistics, including systolic BP, diastolic BP, and pulse rate, were measured using a Sinocare BSX 516 Arm-Type Electronic Blood Pressure Monitor (Changsha Sinocare, Guyuan, China). The average of three measurements was computed.

### 2.9. Fasting Blood Collection and Processing

All participants were instructed to fast for at least eight hours and to refrain from food but could drink water from 8 pm the night before sample collection. Licensed Medical Laboratory Scientists from the Allied Health Profession Council of Ghana collected blood specimens using standard venipuncture techniques and a vacuum blood collection system. The collection process involved collecting 3 mL of fasting blood in a sodium fluoride (NAF) tube. Plasma samples were separated from whole blood by centrifugation, and the diagnosis of diabetes and prediabetes was performed using fasting plasma glucose (FPG)-based on American Diabetes Association (ADA) and International Diabetes Federation (IDF) criteria [2,14]. Participants were classified as normal if their glucose level was below 5.6 mmol/L (100 mg/dL), prediabetes if their FPG was between 5.6 mmol/L and 6.9 mmol/L (100 mg/dL and 125 mg/dL), and diabetes if their FPG level was 7.0 mmol/L or higher (126 mg/dL or higher).

### 2.10. Data Analysis

The study utilized STATA 16 software (StataCorp, College Station, TX, USA) for the statistical analysis. Descriptive statistics, including frequency and percentages, and summary statistics, such as means and standard deviations, were used to assess the prevalence of undiagnosed diabetes based on fasting plasma glucose levels across different variables. Pearson’s chi-square and Cramer’s V statistics were used in bivariate analysis to explore associations between the outcome variable and independent variables. Multivariate regression was conducted using a GSEM [15,16,17] to examine the impact of key biomedical parameters on diabetes status while accounting for relevant variables in the biopsychosocial context. A Gaussian family with an identity link function was applied using the continuous variables in the GSEM method. The over-identified model was retained while following minimum information standards. The final model selection was based on the statistical and theoretical significance of the path coefficients and adherence to minimum information conditions. The direct impact of exogenous variables on their corresponding dependent variables is depicted by black single-headed arrows and the correlation between indirect effects is shown by blue single-headed arrows in Figure 1. All statistically significant effects were assumed to be 5% (*p* < 0.05) at a 95% confidence interval. Model fit was assessed using Likelihood Ratio tests across models as well as comparisons of the Akaike Information Criterion (AIC) and Bayesian Information Criterion (BIC) across models.

## 3. Results 

### 3.1. Descriptive Statistics

#### Sociodemographic Characteristics and Undiagnosed Diabetes Status

A total of 1200 participants were included in this study. Of these, 981 (81.75%) had normal fasting plasma glucose levels, 173 (14.2%) had prediabetes, and 46 (3.83%) had diabetes (Table 1). As shown in Figure 2, a higher prevalence of both prediabetes and diabetes was observed among females than among males, with 122 (15.33%) females having prediabetes, 41 (5.15%) females diagnosed with diabetes, and 633 (79.52%) females displaying normal fasting plasma glucose levels. Among males, only 51 (12.62%) had prediabetes, 5 (1.24%) had diabetes, and 348 (86.14%) had normal fasting plasma glucose levels. Moreover, this study found disparities in the prevalence of undiagnosed diabetes across rural, peri-urban, and urban areas. In rural areas, 21.95% were diagnosed with prediabetes, 6.40% had diabetes, and 71.61% had normal glucose levels. In the peri-urban areas, 13.30% had prediabetes, 2.14% had diabetes, and 84.56% had normal fasting plasma glucose levels. In urban areas, 87.61% of participants had normal fasting plasma glucose levels, 7.61% had prediabetes, and 4.56% had undiagnosed diabetes. These findings indicate that the prevalence of undiagnosed diabetes remains high among rural populations, as shown in Figure 2, Figure 3 and Figure 4.

Fasting plasma glucose levels indicated varied prevalence rates for diabetes and prediabetes among the 13 communities within the Cape Coast Metropolis. Effutu had the highest prevalence of diabetes (12.90%) followed by Abakam (7.69%), Abura (6.59%), and Effutu Mampong (5.56%). Effutu Mampong also had the highest prediabetes prevalence (32.08%), followed by Effutu (29.03%), Ankaful (28.74%), and Ekon (20.31%). Notably, participants from the University Cape Coast Campus, Kotokuraba, and Siwdu had the lowest prevalence rates of both diabetes and prediabetes, whereas normal fasting plasma glucose levels varied across communities, with detailed trends shown in the Figures.

### 3.2. Association between Fasting Plasma Glucose and Independent Variables

Table 2 shows the results of Pearson’s chi-square and Cramer’s V tests, which were used to explore the relationship between the independent variables and undiagnosed diabetes status. Pearson’s chi-square test, a nonparametric method, was employed to investigate the associations between categorical variables in the study and undiagnosed diabetes status. Cramer’s V test was used to evaluate the strength of these associations, with values ranging from 0 to 1. This test provides a standardized measure of association that goes beyond the simple Pearson’s chi-squared statistic, making it a useful tool for assessing the relationship between variables.

Cramer’s V values can vary in interpretation based on context, but typically ≤0.1 signifies a weak association, 0.1–0.3 shows a moderate association, and greater than 0.3 indicates a strong association. The results in Table 2 show that sex, age, type of residence, childhood overweight, and family history of diabetes were directly linked to prediabetes and diabetes status based on fasting plasma glucose levels, with Pearson’s chi-squared indicating that these associations were significant at the 5% level (*p* < 0.05). However, Cramer’s V statistics revealed varying strengths of association, with moderate strength (0.1–0.3) observed for some factors. Conversely, religious affiliation, education level, income status, and alcohol consumption were not significantly associated with prediabetes or diabetes status based on FPG levels. 

### 3.3. Relationships between Cardiometabolic Parameters by Fasting Plasma Glucose

This section presents the distribution of the cardiometabolic parameters across individuals diagnosed with either normal, prediabetic, and diabetes based on the (FPG) levels. The results are presented in means and standard deviations with their significance levels displayed in terms of *p*-values from the one-way analysis of variance (ANOVA). Given that ANOVA determines if there are overall differences among groups, but not which groups differ, we further conducted a Tukey HSD (Honestly Significant Difference) test as a crucial post-estimation test to pinpoint which specific mean differences are statistically significant. The Tukey HSD test compared all the possible pairs of means, helping identify which groups exhibit significant differences and which are similar. This ensures precise insights into group distinctions, vital for drawing accurate conclusions from the data. Based on this, the results are shown in Table 3 where the mean values with different superscripts (alphabets) and asterisks are significantly different. As displayed in Table 3, the results showed that as the participants’ glucose levels increased, their body mass index (BMI), average waist circumferences (WC), and systolic blood pressure (SBP) status worsened as well, while triglycerides (TG) also rose significantly. Notably, the Gamm–Glutamyl Transferase (GGT levels were significantly higher in the participants diagnosed with diabetes based on the FPG levels, potentially indicating liver involvement. Although total cholesterol (TC) and high-density lipoprotein (HDL) levels did not differ significantly across groups, a trend suggested higher TC levels with participants diagnosed with diabetes status. These findings underscore the complex relationship between diabetes status and cardiometabolic health, emphasizing the importance of early intervention and lifestyle modifications to prevent metabolic disorders. The diabetes group had high BMI (29.67 ± 8.48, *p* < 0.001), WC (95.96 ± 10.28, *p* < 0.001), SBP (148.07 ± 24.19, *p* < 0.001), TG (125.47 ± 64.34, *p* < 0.001), GGT (73.55 ± 123.66, *p* > 0.05), TC (183.45 ± 50.90, *p* > 0.05), and low levels of HDL (51.48 ± 19.08, *p* > 0.05). The prediabetes group had similar values for BMI, WC, SPB, and HDL, but lower values for TG and GGT. These findings indicate that individuals with diabetes and prediabetes had higher levels of cardiometabolic parameters than those with normal diabetes status.

### 3.4. Predictive Factors of Fasting Plasma Glucose Levels of the Respondents 

The study used fasting plasma glucose level as an indicator to assess the prevalence of undiagnosed diabetes as an outcome variable in adults aged 25–70 in the Cape Coast Metropolis. The results of the multivariate analysis showed an association between the independent variables, and the GSEM was employed to predict the direct and indirect factors. The Akaike and Bayesian Information Criteria (AIC/BIC) are crucial for assessing model performance. A lower AIC/BIC ratio indicates a better model fit. The proposed regression model with 1200 observations shows substantial improvement over the null model, as indicated by its significantly negative log-likelihood value. We observed that the statistical fit, measured through Likelihood ratio tests, AIC, and BIC, became better. This statistical technique is effective for assessing multiple predictive factors that are directly or indirectly linked to chronic diseases, such as diabetes.

Table 4 displays the GSEM results with coefficients, robust standard errors, probability values, and confidence intervals for direct and indirect pathways. Seven independent variables were used: WC, BMI, TG level, SBP, GGT level, sex, and family history. The dependent variable was diabetes status based on FPG levels. For the indirect effect phase, 15 independent variables were used: TC, TG, HDL, alcohol consumption, childhood overweight, family history, fruit intake, sex, age, socioeconomic status, physical activity, occupation, education, residence, and religion. BMI, TG, SBP, and GGT levels were endogenous variables.

At the 5% significance level, BMI and TG were the dominant predictors of diabetes, with all variables, except WC, demonstrating significant direct associations with FPG. Female sex and a family history of diabetes also directly influenced FPG levels, whereas childhood overweight, family history of diabetes, and female sex increased the predictive pathways involving WC and BMI. Sociodemographic factors such as age, socioeconomic status, and occupation also played an indirect role in affecting FPG levels through WC and BMI. Physical activity appears to have a negative impact on FPG levels, implying a protective effect. 

This study also revealed the complex nature of FPG, influenced by various physiological, lifestyle, and demographic factors. Specifically, the direct hypothesized path showed significant positive effects of BMI (β = 0.322, *p* < 0.01), TG (β = 0.078, *p* < 0.001), SPB (β = 0.084, *p* < 0.001), and GGT (β = 0.027, *p* < 0.01) on FPG levels. Additionally, women (β = 3.640, *p* < 0.05) were more likely to have higher FPG levels than men. WC did not show a significant relationship with FPG but a family history of diabetes had a marginal effect, indicating a nuanced relationship between them. This comprehensive analysis provides valuable insights into the direct predictive factors of diabetes and offers guidance for its prevention and treatment.

The indirect hypothesized path showed a significant positive association between WC and childhood overweight (β = 7.116, *p* < 0.001), family history (β = 3.038, *p* < 0.01), and female sex (β = 8.616, *p* < 0.001). Age also had a significant positive association with WC in middle-aged (β = 3.688, *p* < 0.01) and older adults (β = 4.107, *p* < 0.001). Furthermore, socioeconomic status had a significant positive effect on WC for those in the moderate (β = 2.035, *p* < 0.01) and high (β = 2.035, *p* < 0.01) socioeconomic status.

A negative association between BMI and HDL was observed (β = −0.047, *p* < 0.001), suggesting that higher HDL levels are associated with lower BMI which is protective against diabetes. TC (β = 0.020, *p* < 0.001) displayed a positive association with BMI, indicating that increased total cholesterol leads to higher BMI, which is a risk factor for insulin resistance, a precursor for diabetes. Childhood overweight status (β = 3.876, *p* < 0.001) was a significant predictor, emphasizing the long-lasting impact of early weight experiences on adult health. Physical activity was negatively correlated with BMI, underlining the importance of an active lifestyle for maintaining normal BMI.

Women (β = 3.849, *p* < 0.001) were significantly more likely to be identified to have high BMI than men, and individuals who tended to belong to moderate (β = 1.460, *p* < 0.001) and higher (β = 2.615, *p* < 0.001) socioeconomic status were found to be likely associated with higher BMI levels. Traders (β = 1.653, *p* < 0.01) were found to have a significantly higher BMI than their counterparts. This outcome may be related to long sitting and inactive lifestyle of traders in Ghanaian traditional markets which may increase their risk of developing diabetes.

The association between TG and TC was strong and positive (β = 0.243, *p* < 0.001), implying that total cholesterol levels (higher LDL and low HDL) increase TG levels, which is a risk factor for developing diabetes. Notably, only those aged 56–70 years (β = 13.090, *p* < 0.001) showed a significant and positive association with TG levels, confirming that old age is associated with high TG levels, which is an indirect predictive factor for diabetes. Regarding education, JHS (β = −7.850, *p* < 0.05), SHS (β = −14.067, *p* < 0.01), and tertiary education (β = −14.211, *p* < 0.05) were significantly negatively associated with TG. These educational level results suggest that having a higher education level reduces the likelihood of having higher TG levels and a low likelihood of developing diabetes.

The study also revealed that WC (β = 0.436, *p* < 0.001) was strongly associated with SBP, and TG (β = 0.044, *p* < 0.01) had a positive effect on SBP. However, the influence of sex was minimal as the relationship was only marginal for females. Age had a significant impact on SBP, with higher levels observed in the 35–56 years (β = 7.656, *p* < 0.001) and 56–70 age groups (β = 15.651, *p* < 0.001) compared to young adults. Physical activity was negatively associated with SBP (β = −0.001, *p* < 0.01), whereas peri-urban (β = 4.927, *p* < 0.05) and urban (β = 4.107, *p* < 0.05) residences had a positive impact on SBP. These findings may inform tailored interventions in different settings.

Finally, the study identified alcohol consumption (β = 18.498, *p* < 0.001) as a significant factor that displayed a substantial positive impact on GGT; female sex (β = −17.754, *p* < 0.001) was associated with lower GGT levels than male sex. Additionally, religious affiliations influenced GGT levels, with Christianity (β = −19.771, *p* < 0.05) and Islam (β = −21.626, *p* < 0.05) displaying lower associations compared to belonging to African traditional religion.

## 4. Discussion

Diabetes is a significant global health problem, as reported by the IDF. In Ghana, recent demographic and health surveys (DHS) indicate an increase in obesity and hypertension, which contributes to the worsening of the diabetes situation [20]. Additionally, a lack of healthcare facilities and screening opportunities as well as the high number of undiagnosed cases exacerbate this issue. The study found a 3.8% prevalence of diabetes among the population with a significant prevalence among females in the communities. Delving deep into factors influencing undiagnosed diabetes among the indigenous population, we employed GSEM to investigate direct and indirect factors. Our study differs from previous research in Ghana [21,22,23,24,25], which employed statistical models that did not provide comprehensive explanations of the direct and indirect predictors for diabetes. Our study used a GSEM with path analysis to examine the direct and indirect effects of factors associated with diabetes to inform targeted interventions and public health strategies.

In the GSEM, the findings showed that high TG, SBP, GGT, and female sex directly impact undiagnosed diabetes diagnosed by FPG criteria. Our study aligns with the previous research by Al-Harrasi et al. [26], Farhadipour et al. [18], and Tripathy et al. [19] identified elevated BMI, high TG levels, high SBP, and female sex as the independent predictive factors for diabetes. The findings mean that TG, SBP, and GGT may increase the development of diabetes among this population, especially among the female sex. Clinically, reducing these essential parameters among the population would significantly decrease the development of diabetes among this important population. The high prevalence of diabetes among the female sex could be due to age, as pre-menopausal and menopausal women have metabolic changes due to the drop in estrogen. Also, this finding of women with a higher percentage of diabetes could be due to more women in the sample. Therefore, management of GGT levels is crucial to reduce risk among individuals newly diagnosed with prediabetes and diabetes to regulate their metabolic systems. However, our findings differ from those of Nano et al. [27], as WC and a family history of diabetes did not have a significant impact on diabetes. The WC not contributing to diabetes could be that the measurement criteria are not African-specific; therefore, there is a need for African-specific diagnostic criteria for body sizes. For effective preventive measures, public health strategies and personalized medical interventions should be tailored to the unique characteristics of the population.

We also investigated WC, BMI, TG, SBP, and GGT levels to identify factors that may indirectly increase or decrease the prevalence of diabetes. Our findings indicated that childhood overweight, family history of diabetes, sex, age, and socioeconomic status significantly affected WC. This means that family history and childhood overweight can contribute to increased body size during adulthood as well as increased socioeconomic status. These factors are significant and must be controlled to reduce the prevalence of diabetes among the population. Specifically, women, middle-aged (35–56) and older adults (56–70), and those in the moderate and high socioeconomic status groups had a positive contribution to increased WC, which suggests an indirect positive association with diabetes. These findings are consistent with those of a previous study [28,29,30,31,32]. Therefore, targeted interventions should address these factors in order to reduce the incidence of increased WC as a major factor for diabetes.

Our study identified various factors that impact BMI, including HDL cholesterol levels, childhood obesity, physical activity, sex, socioeconomic status, and occupation. More importantly, we found a negative association between HDL and BMI, suggesting that higher HDL levels are associated with a lower BMI [21,33] significantly serving as a protective factor against insulin resistance. Additionally, we observed a strong negative association between physical activity and BMI, which aligns with the results of previous studies [32,34,35]. However, we also discovered a positive association between BMI and total cholesterol levels, which is in contrast to other studies [36,37].

TG was strongly associated with total cholesterol, especially in individuals consuming fatty diets. Additionally, age was found to have a significant impact on TG levels, with those aged 56–70 years having higher levels of TG. Education level was also found to be a determinant of TG levels, and higher education was associated with lower TG levels. These findings are consistent with those of previous studies by Ahmed et al. [38], Anto et al. [39], Chen et al. [40], Duran et al. [41], and Liu et al. [42]. Therefore, tailored interventions that consider predictors of diabetes are necessary to reduce triglyceride levels and minimize health risks.

This study found significant effects of WC, TG level, age, physical activity, and residence on SBP. WC and TG, in particular, positively influenced SBP, indicating that higher levels of WC and TG increase SBP. These findings align with those of previous studies such as those of Gopinath et al. [43] and Lu et al. [44]. To reduce the SBP, it is important to maintain normal WC and TG levels. Moreover, middle-aged and older adults are more likely to experience a high SBP, which is a predictor for diabetes [23,45]. In contrast, physical activity is negatively associated with SBP, indicating that those who engage in moderate or high levels of physical activity have lower SBP [46,47,48,49,50,51]. These findings underscore the need for targeted interventions to monitor the effects of these determinants on SBP regularly, particularly in the context of controlling diabetes.

Finally, this study investigated the associations between alcohol consumption, sex, religion, and GGT levels. The findings revealed that alcohol intake was significantly associated with higher GGT levels, a predictor of diabetes. Women demonstrated a negative association with GGT levels, suggesting that they were more likely to have lower levels than men. Moreover, religion played a significant role in GGT levels, with Christians and Muslims displaying different negative associations compared to the traditional religious group. Muslims were more likely to have lower GGT levels, while the traditional group had increased levels of high GGT levels and diabetes. These findings offer valuable guidance for public health interventions and personalized care strategies on the reduction in alcohol consumption particularly among men and those who practice traditional religion.

### Strengths and Limitations

This study is the first in Ghana to use a GSEM to identify direct and indirect factors associated with undiagnosed diabetes based on FPG using a reasonably large sample size from a community-based survey. Despite this, this study has a few limitations that are worth mentioning for further research. Notably, the data were obtained from a cross-sectional survey instead of an experimental approach, which implies that the study’s results show an association between the variables, rather than causality. Therefore, we suggest that future studies use an experimental design. Notwithstanding this, the findings of this study are reliable and provide a solid foundation for policy development.

## 5. Conclusions

Undiagnosed diabetes is a significant public health concern in Ghana, necessitating a comprehensive model that accounts for demographic, socioeconomic, clinical, physical activity, and behavioral factors. Although several studies have been conducted on diabetes in Ghana, none have used a GSEM to investigate the direct and indirect associated factors for undiagnosed diabetes. Unlike previous studies, this study applied GSEM to gain insights into direct and indirect factors as modifiable pathways that can help public health professionals prevent diabetes. The results revealed several direct factors associated with diabetes, including waist circumference, BMI, TG, SBP, and GGT. Additionally, sociodemographic, socioeconomic, and lifestyle factors indirectly influence diabetes status. Specifically, women, middle-aged and older adults, those with moderate-to-high socioeconomic status, low physical activity, high alcohol consumption, low education level, rural residence, traditional religion, and certain clinical variables such as HDL and total cholesterol should be monitored to prevent and control diabetes. Effective management of diabetes requires attention to these factors.

## Figures and Tables

**Figure 1 ijerph-21-00836-f001:**
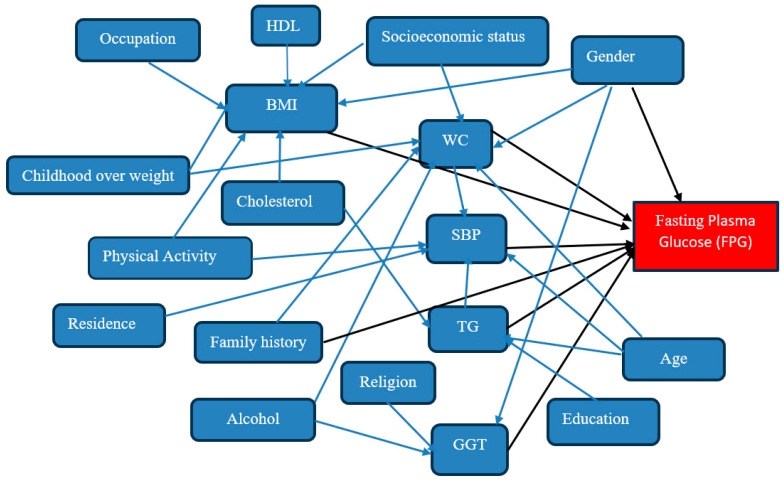
Modified conceptual model according to the risk factors for diabetes for the GSEM analysis. Source: Adapted from (Farhadipour et al. [18]; Tripathy et al. [19]). Note: BMI = Body Mass Index; WC = Waist Conference; TG = Triglyceride; SBP = Systolic Blood Pressure; HDL = Gamma-Glutamyl Transferase (GGT); HDL = High-density Lipoprotein.

**Figure 2 ijerph-21-00836-f002:**
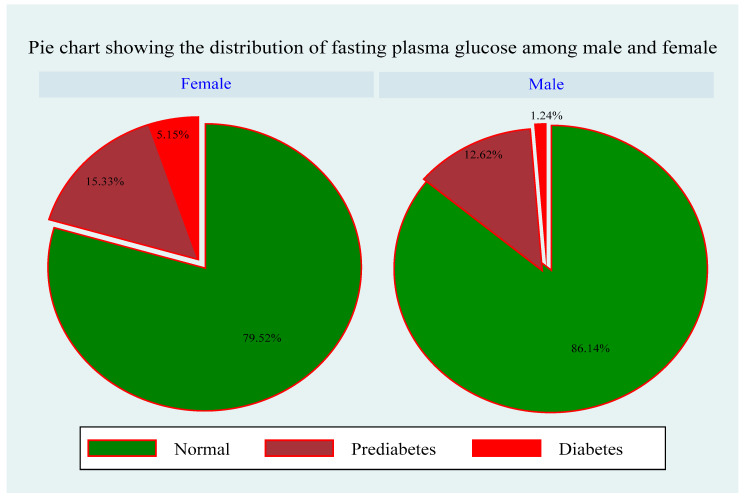
Gender difference in the fasting plasma glucose.

**Figure 3 ijerph-21-00836-f003:**
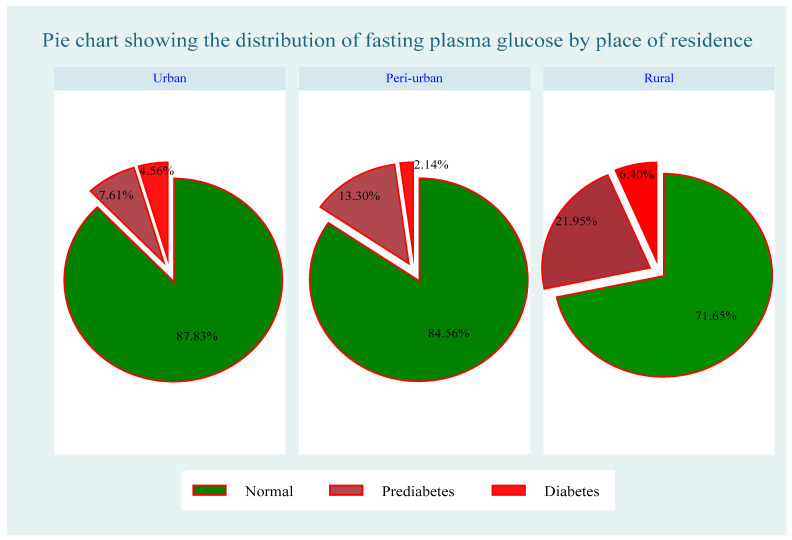
Distribution of fasting plasma glucose across type of place of residence.

**Figure 4 ijerph-21-00836-f004:**
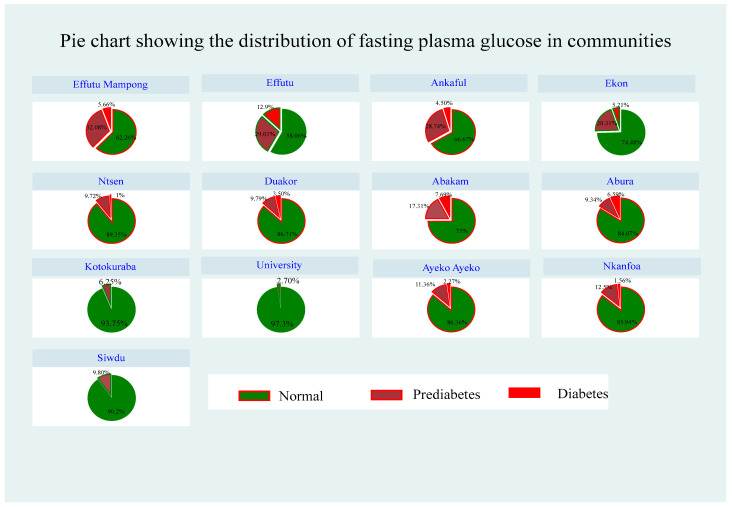
Distribution of fasting plasma glucose in the communities.

**Table 1 ijerph-21-00836-t001:** Prevalence of fasting plasma glucose status of respondents.

Variable	Frequency	Percent (%)
Normal	981	81.75
Prediabetes	173	14.42
Diabetes	46	3.83
All participants	1200	100

**Table 2 ijerph-21-00836-t002:** Distribution of fasting plasma glucose by predictor variables.

Variable	N				Inferential Statistics
		Normal	Prediabetes	Diabetes	
All Participants	1200	731 (60.92)	310 (25.83)	159 (13.25)	n/a
**Sex**					Pearson chi2 (2) = 13.4978 *p*-value = 0.001 Cramer’s V = 0.1061
Male	404	348 (86.14)	51 (12.62)	5 (1.24)	
Female	796	633 (79.52)	122 (15.33)	41 (5.15)	
**Age**					Pearson chi2 (4) = 19.3354 *p*-value = 0.001 Cramer’s V = 0.0898
Young Adult	265	239 (90.19)	22 (8.30)	4 (1.51)	
Middle-aged Adult	594	476 (80.13)	96 (16.16)	22 (3.70)	
Older-aged Adult	341	266 (78.01)	55 (16.13)	20 (5.87)	
**Residence type**					Pearson chi2 (4) = 38.3809 *p*-value < 0.001 Cramer’s V = 0.1265
Urban	263	231 (87.83)	20 (17.60)	12 (4.56)	
Peri-urban	609	515 (84.56)	81 (13.30)	13 (2.13)	
Rural	328	235 (71.65)	72 (21.95)	21 (6.40)	
**Religion**					Pearson chi2 (4) = 3.3254 *p*-value = 0.505 Cramer’s V = 0.0372
Christianity	981	796 (81.14)	143 (14.58)	42 (4.28)	
Islam	136	114 (83.82)	19 (13.97)	3 (2.21)	
Traditional	83	71 (85.54)	11 (13.25)	1(1.20)	
**Education**					Pearson chi2 (8) = 11.9545 Price value = 0.153 Cramer’s V = 0.0706
No formal education	333	262 (78.68)	58 (17.42)	13 (3.90)	
Primary	224	177 (79.02)	34 (15.18)	13 (5.80)	
JHS	431	358 (83.06)	56 (12.99)	17 (3.94)	
SHS	137	116 (84.67)	18 (13.14)	3 (2.19)	
Higher	75	68 (90.67)	7 (9.33)	0 (0.00)	
**Occupation**					Pearson chi2 (16) = 46.8857 *p*-value < 0.001 Cramer’s V = 0.1398
Unemployment	151	121 (80.13)	24 (15.89)	6 (3.97)	
Farmer	76	48 (63.16)	21 (27.63)	7 (9.21)	
Government work	70	66 (94.29)	4 (5.71)	0 (0.00)	
Trader	442	359 (81.22)	64 (14.48)	19 (4.30)	
Driver	21	18 (85.71)	2 (9.52)	1 (4.76)	
Student	6	4 (66.67)	2 (33.33)	0 (0.00)	
Artisan	230	202 (87.12)	25 (10.87)	3 (1.30)	
Fishing	179	147 (82.12)	26 (14.53)	6 (3.35)	
Retiree	25	16 (64.00)	5 (20.00)	4 (16.00)	
**Income status**					Pearson chi2 (4) = 9.2189 *p*-value = 0.056 Cramer’s V = 0.0620
Low	838	453 (79.55)	90 (15.46)	29 (4.98)	
Middle	325	386 (84.28)	61 (13.32)	11 (2.40)	
High	37	132 (82.50)	22 (13.32)	6 (3.75)	
**Alcohol drink**					Pearson chi2 (2) = 3.0918 *p*-value = 0.213 Cramer’s V = 0.0508
No	838	675 (80.55)	127 (15.16)	36 (4.30)	
Yes	362	306 (80.55)	46 (12.71)	10 (2.76)	
**Overweight during childhood**					Pearson chi2 (2) = 12.2665 *p*-value = 0.002 Cramer’s V = 0.1011
No	915	763 (83.39	126 (13.77)	26 (2.84)	
Yes	285	218 (76.49)	47 (16.49)	20 (7.02)	
**History of family diabetes**					Pearson chi2 (2) = 7.8158 *p*-value = 0.020 Cramer’s V = 0.0807
No	1028	848 (82.49)	147 (14.30)	33 (3.21)	
Yes	172	133 (77.33)	26 (15.12)	13 (7.56)	

**Table 3 ijerph-21-00836-t003:** Distribution of cardiometabolic parameters by fasting plasma glucose.

Variable	Total	Normal (731)	Prediabetes (310)	Diabetes (159)	*p*-Value
	N	Mean Values	Mean Values	Mean Values	
BMI (kg/m^2^)	1200	25.67 ± 6.32 ^a^	27.30 ± 8.22 ^b^	29.67 ± 8.48 ^ac^*	<0.001
Average Waist (cm)	1200	88.12 ± 11.70 ^a^	91.56 ± 12.59 ^bc^*	95.96 ± 10.28 ^ac^*	<0.001
Systolic Pressure (mmHg)	1200	132.70 ± 24.09 ^a^	139.31 ± 27.39 ^bc^*	148.07 ± 24.19 ^ac^	<0.001
Triglycerides (mg/dL)	1200	87.03 ± 43.98 ^a^	103.02 ± 48.99 ^bc^*	125.47 ± 64.34 ^ac^*	<0.001
Gamma GT (IU/L)	1200	41.20 ± 59.67 ^a^	46.30 ± 64.26 ^b^	73.55 ± 123.66 ^ac^*	0.053
Total Cholesterol (mg/dL)	1200	170.09 ± 41.29 ^a^	176.93 ± 47.61 ^b^	183.45 ± 50.90 ^b^	0.153
HDL (mg/dL)	1200	51.67 ± 18.74 ^a^	55.50 ± 22.59 ^b^	51.48 ± 19.08 ^c^	0.055

**Note:** Mean values with different superscripts (alphabets) and asterisk are significantly different.

**Table 4 ijerph-21-00836-t004:** Generalized structural equation model (GSEM) regression showing results of risk factors of diabetes based on the fasting plasma glucose levels.

Variables	Coefficient	Std. Err.	Z	*p*-Value	[95% Conf. Interval]	
					Lower	Upper
**Direct Effects** **FPG (outcome var)**						
WC	−0.028	0.072	−0.390	0.699	−0.169	0.113
BMI	0.322	0.123	2.610	0.009	0.081	0.564
TG	0.078	0.014	5.560	<0.001	0.051	0.106
SBP	0.084	0.027	3.160	0.002	0.032	0.136
GGT	0.027	0.010	2.610	0.009	0.007	0.047
Gender						
Female	3.640	1.465	2.490	0.013	0.769	6.511
Family has diabetes (No)						
Yes	3.381	1.830	1.850	0.065	−0.205	6.967
**Indirect Effects** **WC (outcome var)**						
Drinking of alcohol (Ref = No)						
Yes	1.358	0.713	1.910	0.057	−0.039	2.755
Childhood overweight (Ref = No)						
Yes	7.116	0.727	9.790	<0.001	5.691	8.541
Family history of diabetes (Ref = No)						
Yes	3.038	0.879	3.460	0.001	1.316	4.760
Fruit intake	−0.119	0.606	−0.200	0.844	−1.306	1.068
Gender (Ref = Male)						
Female	8.616	0.720	11.970	<0.001	7.205	10.027
Age (Ref = 25–34)						
35–55	3.688	0.781	4.720	<0.001	2.157	5.219
56–70	4.107	0.866	4.740	<0.001	2.409	5.804
Socioeconomic status (Ref = Low)						
Moderate	2.035	0.671	3.030	0.002	0.719	3.351
High	2.970	0.987	3.010	0.003	1.036	4.904
**BMI (outcome var.)**						
HDL	−0.047	0.009	−5.170	<0.001	−0.065	−0.029
Total Cholesterol	0.020	0.004	4.780	<0.001	0.012	0.028
Childhood overweight (Ref = No)						
Yes	3.876	0.413	9.390	<0.001	3.067	4.686
Physical activity	−0.000	0.000	−2.120	0.034	−0.000	−0.000
Fruit intake	−0.009	0.339	−0.030	0.979	−0.673	0.654
Gender (Ref = Male)						
Female	3.849	0.452	8.510	< 0.001	2.963	4.736
Socioeconomic status (Ref = Low)						
Moderate	1.460	0.391	3.730	<0.001	0.693	2.226
High	2.618	0.579	4.520	<0.001	1.482	3.753
Occupation (Ref = Unemployed)						
Farmer	−1.345	0.837	−1.610	0.108	−2.985	0.295
Gov’t employee	1.475	0.919	1.610	0.108	−0.326	3.276
Trader	1.653	0.577	2.860	0.004	0.522	2.785
Driver	0.541	1.415	0.380	0.702	−2.232	3.315
Student	−0.382	2.454	−0.160	0.876	−5.192	4.428
Artisan	0.347	0.668	0.520	0.603	−0.961	1.655
Fisherfolk	0.807	0.682	1.180	0.236	−0.529	2.143
Retiree	−0.599	1.293	−0.460	0.643	−3.134	1.936
**TG (outcome var.)**						
Total Cholesterol	0.243	0.030	8.040	<0.001	0.184	0.302
Age (Ref = Young-age < 35)						
Middle-Age adult (35–55)	0.997	3.426	0.290	0.771	−5.717	7.711
Old-age adult (56–70)	13.090	3.858	3.390	0.001	5.529	20.652
Education (Ref = No education)						
Primary	0.990	3.834	0.260	0.796	−6.524	8.504
JHS	−7.850	3.287	−2.390	0.017	−14.291	−1.408
SHS	−14.067	4.717	−2.980	0.003	−23.313	−4.822
Tertiary	−14.211	5.762	−2.470	0.014	−25.506	−2.917
**Systolic (outcome var.)**						
WC	0.436	0.061	7.180	<0.001	0.317	0.555
TG	0.044	0.015	2.980	0.003	0.015	0.073
Gender (Ref = Male)						
Female	0.422	1.515	0.280	0.781	−2.549	3.392
Age (Ref = Young adult < 35)						
Middle-age adult (35–55)	7.656	1.719	4.450	<0.001	4.286	11.026
Old-age Adult (56–70)	15.651	1.928	8.120	<0.001	11.872	19.430
Physical activity	−0.001	0.000	−2.910	0.004	−0.001	−0.000
Residence (Ref = Rural)						
Peri-urban	4.927	1.924	2.560	0.010	1.156	8.697
Urban	4.107	1.583	2.590	0.009	1.004	7.210
**GGT (outcome var.)**						
Drink Alcohol (Ref = No)						
Yes	18.498	4.344	4.260	<0.001	9.984	27.011
Gender (Ref = Male)						
Female	−17.754	4.113	−4.320	<0.001	−25.814	−9.693
Religion (Ref = Tradition)						
Christianity	−19.771	7.335	−2.700	0.007	−34.147	−5.396
Islam	−21.626	9.070	−2.380	0.017	−39.404	−3.848
No of Obs	1200					
Goodness of Fit						
AIC	64,394.78					
BIC	64,715.46					
log likelihood	−32,134.39					

## Data Availability

The original contributions presented in the study are included in the article, further inquiries can be directed to the corresponding author.

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
