# Peer review of "Unmasking the Risk Factors Associated with Undiagnosed Diabetes and Prediabetes in Ghana: Insights from Cardiometabolic Risk (CarMeR) Study-APTI Project"

_ijerph, 2024, doi:10.3390/ijerph21070836_

Round 1

Reviewer 1 Report

Comments and Suggestions for Authors

This study explores risk factors for diabetes in Ghana in a cross-sectional study.

In abstract it was not clear whether the participants were people who had never been diagnosed with diabetes or people with diabetes that had not been diagnosed. Assume the former, that would make more sense.

Be careful to define all abbreviations, in particular in abstract what is GGT?

When looking at risk factors for undiagnosed diabetes there are two distinct questions to be considered, which variables predict development of diabetes (obesity, age etc) and which variable predict whether diabetes will be diagnosed (eg other conditions that might make health be more closely monitored).

Why would factors related to managing diabetes be associated with undiagnosed diabetes?

The study is cross-sectional and does not seem designed to identify causal associations, how will non-causal risk factors be useful in prevention, how will they be identified?

Please provide sample size calculations to explain how the number of1200 was determined.

If I understand correctly Figure 1 shows that age is associated with FPG only via TG, SBP and WC. Is this logical based on physiology?

No mention of ethical approval or consent procedure is included.

The first paragraph of results is not results from the study and should be removed.

Line 407-8, it is not surprising that a higher BMI is associated with a lower HDL, higher HDL is better and would be associated with a healthy ‘normal BMI.

Final paragraph should be conclusions not strengths and weaknesses.

The findings are very specific to the study population and the discussion does not help readers to see how the findings could be useful outside this location.

Comments on the Quality of English Language

Although the English reads ok on first glance, the whole paper is difficult to understand which may be due to the writing style.

Author Response

Comments and Suggestions for Authors

This study explores risk factors for diabetes in Ghana in a cross-sectional study.

In abstract it was not clear whether the participants were people who had never been diagnosed with diabetes or people with diabetes that had not been diagnosed. Assume the former, that would make more sense. This is good catch, we have included that statement in the manuscript “1200 adults who perceived themselves as healthy and have not been previously diagnosed with diabetes” This statement makes that aspect of the abstract clearer.

Be careful to define all abbreviations, in particular in abstract what is GGT? The gamma-glutamyl transferase (GGT) has been inputted in the text.

When looking at risk factors for undiagnosed diabetes there are two distinct questions to be considered, which variables predict development of diabetes (obesity, age etc) and which variable predict whether diabetes will be diagnosed (eg other conditions that might make health be more closely monitored). In our research BMI, TG, SBP, gamma-glutamyl transferase (GGT), and female sex directly predicted diabetes among the participants. Such that as BMI, TG, SBP and GGT increased, diabetes increased particularly among the female gender.

Why would factors related to managing diabetes be associated with undiagnosed diabetes? These are factors that increase the risk of being diagnosed with diabetes in the indigenous communities of Ghana. More importantly, the factors identified when controlled among the healthy adult population, diabetes development will be reduced.

The study is cross-sectional and does not seem designed to identify causal associations, how will non-causal risk factors be useful in prevention, how will they be identified? We think the study design used was appropriate to explain a causal relationship between our independent variables used in the model and how they predicted undiagnosed diabetes among the population. And this has been applied other studies ……….

Please provide sample size calculations to explain how the number of1200 was determined. In general, the CarMeR study aimed at a power of 0.90 with α=0.05, incorporating the Bonferroni correction. Using these parameters, a sample size of approximately 1000 participants would be sufficient for the Cape Coast's rural, peri-urban and urban population subsets to detect a 10% proportion of undiagnosed T2D. The parameters used were based on the proportion of Agyemang et al. (2014).

 If I understand correctly Figure 1 shows that age is associated with FPG only via TG, SBP and WC. Is this logical based on physiology? Yes! Age predicted TG, SBP and WC. Indicating that increased age is associated with increased TG, SBP, and WC and together they contributed to increased FPG levels.

No mention of ethical approval or consent procedure is included. The study protocol was approved with ethical clearance from University of Cape Coast Institutional Review Board. This is evidence in page 4 line 154 and 155.

The first paragraph of results is not results from the study and should be removed. The preliminary introduction paragraph to the results presentation has been removed as suggested.

Line 407-8, it is not surprising that a higher BMI is associated with a lower HDL, higher HDL is better and would be associated with a healthy ‘normal BMI. Yes, that support previous research findings.

Final paragraph should be conclusions not strengths and weaknesses. We thank the reviewer for the suggestion but humbly pleaded that the penultimate paragraph indicated the conclusion of the findings. However, per MCPI criteria, strengths and weaknesses needed to be included in the paper. We therefore think that the final paragraph is relevant and important.

The findings are very specific to the study population and the discussion does not help readers to see how the findings could be useful outside this location. The discussion has been improved to enhance the usefulness of the study.

Reviewer 2 Report

Comments and Suggestions for Authors

The paper demonstrates the risk factors for undiagnosed Diabetes and 2 Prediabetes in Ghana, associating this disease with cardiometabolic risk development. The article provides relevant results and new insight for a better understanding of the risk factors involved in diabetes development in the Ghana population. However, I have some concerns outlined in the comments below.

- Please check all acronyms in the paper. The authors need to quote them in full first. Check the entire text.

- In the keywords, please replace risk factors with risk factors for diabetes. This change will increase the chances of the other authors finding the article in searches for diabetes-related data.

- What dietary changes contribute to the increase in diabetes and prediabetes? Please add this information to the introduction on line 92.

- Did the authors consider the glucocorticoids and thiazide diuretics use as exclusion criteria? Because these drugs can cause changes in glycemic levels. Therefore, individuals using these drugs should not be part of the study. Please add detailed information about the medications used by patients and excluded from the study in the materials and methods.

- Please remove the text between lines 220 and 224 from the paper. In the results, authors must only insert a description of the data found in the study.

- The authors found a higher percentage of women with prediabetes and diabetes compared to men. One factor that may contribute to this result is age, as pre-menopausal and menopausal women have metabolic changes due to the drop in estrogen. Therefore, the authors should add data on age by sex and carry out an analysis taking age between sexes into account to correlate with this finding of women with a higher percentage of diabetes. These changes associated with menopause also need to be added to the discussion and conclusion.

Author Response

Comments and Suggestions for Authors

The paper demonstrates the risk factors for undiagnosed Diabetes and 2 Prediabetes in Ghana, associating this disease with cardiometabolic risk development. The article provides relevant results and new insight for a better understanding of the risk factors involved in diabetes development in the Ghana population. However, I have some concerns outlined in the comments below.

- Please check all acronyms in the paper. The authors need to quote them in full first. Check the entire text. We have done these necessary corrections especially for those unknown acronyms.

- In the keywords, please replace risk factors with risk factors for diabetes. This change will increase the chances of the other authors finding the article in searches for diabetes-related data. We have made this change.

- What dietary changes contribute to the increase in diabetes and prediabetes? Please add this information to the introduction on line 92. We have added dietary changes such increased in consumption of energy dense foods, sweets and fast foods contributing increase in the risk factors of diabetes

- Did the authors consider the glucocorticoids and thiazide diuretics use as exclusion criteria? Because these drugs can cause changes in glycemic levels. Therefore, individuals using these drugs should not be part of the study. Please add detailed information about the medications used by patients and excluded from the study in the materials and methods. All those on glucose reducing medicines like metformin, glucocorticoids and thiazide diuretics were excluded from the study and this addition have been included in the manuscript.

- Please remove the text between lines 220 and 224 from the paper. In the results, authors must only insert a description of the data found in the study. The preliminary introduction to the results presentation has been removed as suggested.

- The authors found a higher percentage of women with prediabetes and diabetes compared to men. One factor that may contribute to this result is age, as pre-menopausal and menopausal women have metabolic changes due to the drop in estrogen. Therefore, the authors should add data on age by sex and carry out an analysis taking age between sexes into account to correlate with this finding of women with a higher percentage of diabetes. These changes associated with menopause also need to be added to the discussion and conclusion. We agree that premenopausal and menopausal ages of women could contribute to metabolic changes in women due reduction in estrogen production. We also think that the plausible explanation to this finding could also be attributed more females being in the sample. These explanations have been added to the discussion.

Reviewer 3 Report

Comments and Suggestions for Authors

This is an interesting manuscript with the aim was to evaluate fasting plasma levels, a critical indicator of diabetes, and the associated direct and indirect risk or protective factors. Childhood obesity, physical inactivity, sex, socioeconomic status, and family history indirectly contributed to diabetes development in Ghana.

This manuscript is clear and easy to read. It is presented in a well-structured manner. The cited references are mostly current. There is a reference of self-citation that is under review; I am unsure if it is appropriate to present it in this way.

This manuscript is a valuable contribution to the literature on the prevalence and impact of diabetes in Ghana. In the introduction, the authors emphasize the significance of identifying undiagnosed diabetes in different regions of Ghana, which is a critical step in addressing the growing burden of diabetes in the country. The authors also highlight the existing gap in knowledge for public health policies, which underscores the need for more research in this area.

The authors provide a comprehensive review of the literature on diabetes in Ghana, highlighting the importance of addressing diabetes in Ghana, given the country's rapidly changing demographics and urbanization, which are contributing to a rise in the prevalence of diabetes and other non-communicable diseases.

The experimental design is appropriate to test the hypothesis. The statistical analysis is appropriate for answering the question posed in the study. I have only a few suggestions for a better understanding of the manuscript.

On page 3 of 17, line 145, specify whether individuals on antihypertensive medications were included or not.

In Table 2, the variables Age, residence, etc. should be in the column labeled "Variable."

On page 9 of 17, in Table 3, the units of measurement for each variable should be specified.

On page 9 of 17, line 282, specify the actual p-value for the reported variable comparison.

Page 9 of 17, line 287, please adjust the standard deviation value according to the table, and on line 288, specify the TC value as reported in Table 3.

On page 12 of 17, from lines 335 to 349, please review the beta values to ensure they correspond to those reported in Table 5.

On page 14 of 17, they mention "Surprisingly, we found a negative correlation between HDL and BMI..." but this is widely reported.

The conclusions are consistent with the evidence and arguments presented revealed several direct risk factors associated with diabetes, including waist circumference, body mass index (BMI), triglycerides (TG), systolic blood pressure (SBP), and glutamate-pyruvate transaminase (GGT).

Author Response

Comments and Suggestions for Authors

This is an interesting manuscript with the aim was to evaluate fasting plasma levels, a critical indicator of diabetes, and the associated direct and indirect risk or protective factors. Childhood obesity, physical inactivity, sex, socioeconomic status, and family history indirectly contributed to diabetes development in Ghana.

This manuscript is clear and easy to read. It is presented in a well-structured manner. The cited references are mostly current. There is a reference of self-citation that is under review; I am unsure if it is appropriate to present it in this way. We think, it can be done since the protocol is under review.

This manuscript is a valuable contribution to the literature on the prevalence and impact of diabetes in Ghana. In the introduction, the authors emphasize the significance of identifying undiagnosed diabetes in different regions of Ghana, which is a critical step in addressing the growing burden of diabetes in the country. The authors also highlight the existing gap in knowledge for public health policies, which underscores the need for more research in this area. We appreciate comments of the reviewer.

The authors provide a comprehensive review of the literature on diabetes in Ghana, highlighting the importance of addressing diabetes in Ghana, given the country's rapidly changing demographics and urbanization, which are contributing to a rise in the prevalence of diabetes and other non-communicable diseases. We appreciate the comments of the reviewers.

The experimental design is appropriate to test the hypothesis. The statistical analysis is appropriate for answering the question posed in the study. I have only a few suggestions for a better understanding of the manuscript.

On page 3 of 17, line 145, specify whether individuals on antihypertensive medications were included or not. Those persons on antihypertensive medications were included in the study and this information has been added to the manuscript.

In Table 2, the variables Age, residence, etc. should be in the column labeled "Variable." They have been rightfully placed under the column labeled Variable.

On page 9 of 17, in Table 3, the units of measurement for each variable should be specified. Good catch from the reviewer, the units of measurement have been added and highlighted yellow.

On page 9 of 17, line 282, specify the actual p-value for the reported variable comparison. We appreciate this reviewer’s comment. We have specifically added the p-values and improve on the results presentation.

Page 9 of 17, line 287, please adjust the standard deviation value according to the table, and on line 288, specify the TC value as reported in Table 3. Good catch from the reviewer. The correction has been done and highlighted yellow.

On page 12 of 17, from lines 335 to 349, please review the beta values to ensure they correspond to those reported in Table 5. Excellent observation from the reviewer. We have made the necessary reviews and captured correct beta and significant values in the manuscript.

On page 14 of 17, they mention "Surprisingly, we found a negative correlation between HDL and BMI..." but this is widely reported. We accept this observation have and changed surprisingly to ‘more importantly’.

The conclusions are consistent with the evidence and arguments presented revealed several direct risk factors associated with diabetes, including waist circumference, body mass index (BMI), triglycerides (TG), systolic blood pressure (SBP), and glutamate-pyruvate transaminase (GGT). We appreciate the comments of the reviewer.

Round 2

Reviewer 1 Report

Comments and Suggestions for Authors

As noted previously, this is a cross-sectional study design, which is fine and can identify predictors of undiagnosed diabetes, but the findings cannot be interpreted as causal associations. May be better to talk about predictors rather than risk factors as this work does not suggest that changing levels of these factors would change risk of diabetes or pre-diabetes. Similarly, unless you are testing causal associations you should not use expressions such as ‘: Diabetes in indigenous communities is directly driven by blood lipid, BMI, SBP, and alcohol levels, as well as being female.’ which implies causality. Also does not support that you can develop interventions based on these factors. The whole manuscript needs to be edited to avoid using phrases that imply causality.

How was the number 1200 participants identified?

It is Figure 2, not 1, that shows higher proportion of diabetes and pre-diabetes in females relative to males.

The results of multivariate analysis are not correlation (line 312).

Line 357-8, isn’t it more likely that higher BMI causes highest total cholesterol than the other way around?

Line 363-6 need editing to clarify. Inappropriate use of correlation here.

The statistical associations described do not seem to be based on physiology. Why would high TC cause high TG? More likely they both arise from a common cause.

Line 390, what is ‘a traditional’?

Strengths and limitations should be moved into the discussion somewhere before the conclusion. The conclusion should take the strengths and limitations into account.

The weakness of cross-sectional design for causal inference is acknowledged but the discussion does not really take account of this, treating associations as causal and targets for interventions. The best this work can do is help target people at higher risk of elevated fasting glucose

Comments on the Quality of English Language

Generally the English is reasonable, but correlation is used incorrectly in several instances.

Author Response

Review Report Form

Open Review

Quality of English Language

( ) I am not qualified to assess the quality of English in this paper
( ) English very difficult to understand/incomprehensible
( ) Extensive editing of English language required
(x) Moderate editing of English language required
( ) Minor editing of English language required
( ) English language fine. No issues detected

Yes

Can be improved

Must be improved

Not applicable

Does the introduction provide sufficient background and include all relevant references?

(x)

( )

( )

( )

Are all the cited references relevant to the research?

(x)

( )

( )

( )

Is the research design appropriate?

(x)

( )

( )

( )

Are the methods adequately described?

( )

(x)

( )

( )

Are the results clearly presented?

( )

(x)

( )

( )

Are the conclusions supported by the results?

( )

( )

(x)

( )

Comments and Suggestions for Authors

As noted previously, this is a cross-sectional study design, which is fine and can identify predictors of undiagnosed diabetes, but the findings cannot be interpreted as causal associations. May be better to talk about predictors rather than risk factors as this work does not suggest that changing levels of these factors would change risk of diabetes or pre-diabetes. Similarly, unless you are testing causal associations you should not use expressions such as ‘: Diabetes in indigenous communities is directly driven by blood lipid, BMI, SBP, and alcohol levels, as well as being female.’ which implies causality. Also does not support that you can develop interventions based on these factors. The whole manuscript needs to be edited to avoid using phrases that imply causality. The risk factors used in the manuscript do not connote causality or causal effect but only explaining causal relationship. Causality can only be explained when we indicate that the cause precedes effect. However, some of these phrases in the manuscript have been corrected to reflect associations instead of causality. The corrections have shown highlighted using Green color.

How was the number 1200 participants identified? The participants were identified and recruited from the community after announcement. This information can be seen in lines127-129.                                                                                                 

It is Figure 2, not 1, that shows higher proportion of diabetes and pre-diabetes in females relative to males. Good catch from the reviewer, we have consequently corrected this anomaly.

The results of multivariate analysis are not correlation (line 312). Great observation, we changed the correlation to association

Line 357-8, isn’t it more likely that higher BMI causes highest total cholesterol than the other way around? It is but the results presented is based on data. There is a positive associated between the two variables, one can contribute to the increase in the other.

Line 363-6 need editing to clarify. Inappropriate use of correlation here. We have edited the correlation and replaced it with association in the manuscript.

The statistical associations described do not seem to be based on physiology. Why would high TC cause high TG? More likely they both arise from a common cause. Higher LDL and low HDL can increase TC levels and that will be physiologically correct. The choice of words may have contributed this confusion. We have rephrased that sentence.   

Line 390, what is ‘a traditional’? Traditional here means African traditional religion. We have rephrased that sentence to bring clear understanding.  

Strengths and limitations should be moved into the discussion somewhere before the conclusion. The conclusion should take the strengths and limitations into account. We thank the reviewer for the suggestion, we have incorporated these all important suggestions to increase the scholarliness of the manuscript.

The weakness of cross-sectional design for causal inference is acknowledged but the discussion does not really take account of this, treating associations as causal and targets for interventions. The best this work can do is help target people at higher risk of elevated fasting glucose. We appreciate your review and are eternally grateful to have learnt a lot from your expertise. We have tried our best to work on all the issues raised.

Comments on the Quality of English Language

Generally the English is reasonable, but correlation is used incorrectly in several instances.

Submission Date

26 February 2024

Date of this review

24 Apr 2024 02:44:32